# Relapse Predictors in Antineutrophil Cytoplasmic Antibody (ANCA)-Associated Vasculitis

**DOI:** 10.3390/diagnostics14171849

**Published:** 2024-08-24

**Authors:** Katarzyna Wawrzycka-Adamczyk, Mariusz Korkosz, Jacek Musiał, Krzysztof Wójcik

**Affiliations:** 1Rheumatology, Immunology and Internal Medicine Department, Kraków University Hospital, ul Jakubowskiego 2, 30-688 Kraków, Poland; mariusz.korkosz@uj.edu.pl (M.K.); krzysztof.wojcik@uj.edu.pl (K.W.); 2Faculty of Medicine, Jagiellonian University Medical College, 31-008 Kraków, Poland; jacek.musial@uj.edu.pl

**Keywords:** vasculitis, relapse, predictors

## Abstract

Antineutrophil cytoplasmic antibody (ANCA)-associated vasculitides (AAVs) are a group of rare diseases with a chronic and relapsing course. Recent treatment guidelines offer many therapeutic options depending mainly on the type of diagnosis and disease manifestations. Areas that remain under discussion include whether all patients diagnosed with AAV belong to a homogeneous group with a similar prognosis at baseline or if the type and duration of remission-inducing treatment should depend on factors other than just diagnosis and disease severity. The aim of this review is to present the recent literature on the tools available to use while evaluating the risk of relapse in patients upon presentation as well as potential biomarkers of proceeding flare in patients upon remission.

## 1. Introduction

Antineutrophil cytoplasmic antibody (ANCA)-associated vasculitides (AAVs) are a group of rare diseases with an incidence of 13–20 cases per million per year worldwide, including granulomatosis with polyangiitis (GPA), microscopic polyangiitis (MPA) and eosinophilic GPA (EGPA) [1,2,3,4]. The incidence of AAV has grown markedly over the last 30 years, which might be explained by the availability of ANCA testing, growing awareness of the diseases and evolution of classification criteria [5]. AAV clinical manifestations mostly reflect the fact that inflammation affects small and medium vessels and includes mostly upper and lower respiratory tract involvement, glomerulonephritis, and skin, eye and neurological symptoms. Since their first description in the 1930s, advancements in research on the diagnosis and treatment of these diseases have transformed them from rapidly progressing fatal conditions into chronic ones [6,7]. The latest treatment guidelines offer many therapeutic options depending on the type of diagnosis, disease manifestations, and other clinical variables. The intensity of the initial immunosuppressive remission induction depends mainly on the initial diagnosis of limited or generalized form of the disease, while maintenance therapy intensity and duration is rather uniform [8]. The treatment recommendations provide guidance for therapeutic decisions but do not address all questions regarding the care of patients with AAV. Areas that remain under discussion include whether all patients diagnosed with AAV belong to a homogeneous group with a similar prognosis at baseline or if the type and duration of remission-inducing treatment should depend on factors other than just diagnosis and the differentiation between limited and generalized forms. Given that AAV exacerbations lead to the accrual of organ damage, another research aim is to create a set of diagnostic tools that, along with the form of the disease, could predict the long-term efficacy of the applied treatment or the likelihood and timing of relapse [9]. Another challenge in AAV management is the outpatient monitoring of those who have achieved remission. Commonly used disease activity assessment tools, which guide immunosuppressive treatment intensification, rely on a set of results, including imaging, laboratory tests, and consultation outcomes. To obtain these results, it is necessary to repeat them regularly in asymptomatic patients or in an accelerated manner for patients reporting symptoms. The fact that some clinical manifestations, such as hematuria, may present with few symptoms can delay the recognition of disease relapse, thereby predisposing patients to persistent changes and the accumulation of irreversible organ damage. Some simple sets of laboratory tests are proposed to perform on a regular basis for an early relapse detection in patients upon remission, but other sensitive and specific tools are still under investigation [10,11].

It is worth emphasizing that EGPA is both pathogenetic and clinically different from other AAVs. Patients of this entity are rarely participants of the same clinical trials or other scientific studies assessing the course of exacerbations, so EGPA is not included in the study below.

## 2. Methodology

The methodology of this review was based on the literature search for the MEDLINE database including studies published before 30 May 2024. Only studies written in English were analyzed. The main search strategy included adult GPA, MPA and AAV cohorts combined with relapse terminology.

## 3. Baseline Relapse Risk Evaluation

The clinical manifestation at flare of AAV seems to play a role in predicting future exacerbation. Single-center experiences most commonly pointed at proteinase 3 (PR3) ANCA presence and lung involvement as factors predictive of relapse [12,13]. Large multicenter clinical trials have delivered further data to analyze the set of basic risk factors of relapse in AAV. In 2008, a group of patients in the Glomerular Disease Collaborative Network (GDCN, *n* = 350) in the southeastern US was compared with a group of patients in the French Vasculitis Study Group (FVSG, n = 434). During a mean follow-up of 45 months, the risk of exacerbation in the GDCN group was shown to coexist with the presence of anti-PR3 antibodies (HR 1.68), lung involvement (HR 1.68), and upper respiratory tract involvement (HR 1.58) [14]. Pooled results from NORAM, CYCAZAREM, CYCLOPS and MEPEX (>500 patients) (pooled trials cohort) showed that 38% of patients experienced relapse during the observation period (between 36 and 132 months). Anti-proteinase 3 antibodies and cardiovascular involvement were found to be independently associated with a higher risk of relapse, while a creatinine level more than 200 µmol/L at baseline was strongly associated with a reduced risk of flare [15]. A meta-analysis of 16 trials published in 2021 confirmed that significant risk factors for relapse include the presence of anti-PR3 ANCA positivity [HR 1.69], cardiovascular involvement [HR 1.78], creatinine >200 µmol/L [HR 0.39] and creatinine 101–200 µmol/L [HR 0.81] [16]. 

Similar observations were made in a meta-analysis searching for relapse predictors in cyclophosphamide-treated patients by He et al. A lower level of baseline serum creatine, proteinase 3 (PR3)-ANCA positivity at diagnosis and extrarenal organ involvement (lung, cardiovascular, upper respiratory, and gastrointestinal involvement), were found risk factors for an exacerbation at baseline [17]. A significant role of creatinine level on relapse risk was also confirmed in a big Polish cohort analysis. Also, skin, ENT and eye involvement were found to increase the risk of flare [18].

### 3.1. ANCA Testing

ANCA antibodies are an important pathogenetic element in the development of vasculitis with high sensitivity and specificity for AAV diagnosis [19,20]. Actual guidelines suggest using one-step antibody testing as a rapid, non-invasive tool in the diagnosis of patients with suspected vasculitis, based on the EUVAS study comparing the performance of ANCA IF and immunoassays and the subsequent changes introduced by the revised 2017 international consensus on the testing of ANCAs in granulomatosis with polyangiitis and microscopic polyangiitis [21,22]. Their identification is important in the process of classifying patients with a histopathological diagnosis of small vessel vasculitis [23,24,25]. However, it should be noticed, on the one hand, that some patients diagnosed with GPA, MPA and EGPA remain seronegative, and on the other hand, ANCA positivity is observed also in other diseases such as inflammatory bowel disease or drug-induced [26,27,28]. It is also worth emphasizing that racial differences are observed in the distribution of AAV serotypes in various populations, e.g., Caucasian patients diagnosed with GPA are mainly anti-ANCA-PR3-positive (65–75%), while in the Chinese population, the majority of patients with GPA show myeloperoxidase (MPO) positivity (71%) [29,30]. Overall, it should be concluded that while ANCA testing is an indispensable diagnostic tool in suspected AAV and has an important role in the classification process, it also has some limitations [31]. The performance of ANCA testing, its type and titer in predicting the recurrence of relapsing AAV patients remains debatable. The first reports regarding the potential role of ANCA titer in predicting exacerbations date back to the 1990s. The Kyndt team monitored disease status over 22 months in groups of patients completing remission-inducing treatment for GPA (then Wegener’s granulomatosis, n = 21) and MPA (n = 17). They observed that the baseline ANCA pattern (IF) did not differentiate relapsers from non-relapsers; however, relapses occurred more frequently in patients with initial lung involvement [16 of 23 (70%) vs. 6 of 20 (30%)] and persisting ANCA presence (86% vs. 20% in the ANCA-negative group) [32]. The following years brought further evidence of the usefulness of dividing patients with AAV in terms of antibody specificity in stratifying the risk of exacerbations [33,34,35,36]. Lionaki’s group compared this stratification system to the disease classification criteria and Chapel Hill Consensus Conference (CHCC) definitions, indicating that ANCA’s specificity had the best relapse predictive model fit of all the tested methods [37]. Similar observations were provided by the first large clinical trials comparing different treatment regimens to induce and maintain remission in AAV. A cluster analysis of the results of five such studies (n = 673) selected the following clusters: ‘renal AAV with PR3-ANCA positivity (40% of subjects), ‘renal AAV without PR3-ANCA’ (32%), ‘non-renal AAV’ (12%), ‘cardiovascular AAV’ (9%) and ‘gastrointestinal AAV’ (7%), differing from each other in both overall survival and the percentage of patients experiencing disease relapse, indicating the first subgroup as having the highest risk of recurrence [38]. In 2013, a retrospective analysis was conducted on factors responsible for not achieving disease remission within 6 months in the RAVE study (CYC/AZA vs. RTX in AAV). Among the study participants, 19% experienced disease relapse after initial improvement but before the 6-month mark of induction treatment. Among those who had severe flares, 92% were PR3-ANCA-positive, and 83% had GPA. A history of disease exacerbations prior to starting induction therapy was also considered an additional risk factor for early disease relapse at that time [39]. Not all analyses regarding the relationship between antibody type and disease relapse have been consistent. In a Japanese study from 2015, it was shown that both ANCA-PR3+ GPA and ANCA-MPO+GPA patient groups relapsed more frequently than MPA patients. It is worth noting, however, that this study involved a smaller group of patients compared to other studies on this topic and focused on a Japanese population characterized by significant differences in the type of antibodies present in GPA compared to Caucasian populations [40]. A pooled analysis from two large RCTs, RAVE and WGENT, found in 2016 that the risk of exacerbations in the MPO-ANCA-positive GPA and PR3-positive GPA groups did not differ from each other, but in the former group, it was higher than in the MPO-positive MPA group at both 12 and 18 months of follow-up. The risk of relapse in this cohort study was associated more closely with disease type rather than ANCA type [41]. A similar observation was made from a retrospective analysis of patients from Caen University Hospital (n = 150). Relapse-free survival was shorter in the group of patients with GPA than for patients with MPA; however, no differences in the risk of exacerbation were observed between patients with PR3+ and MPO+ [42]. The differences between the findings of these studies and earlier reports may stem from several factors. Firstly, the analyzed groups differ demographically and clinically. Some study populations include participants from clinical trials, which could impact the generalizability of findings to broader patient populations. Additionally, there are significant differences in cohort creation methodologies, such as the proportion of patients with new diagnoses versus those experiencing disease exacerbations. Moreover, variations exist in the speciality of medical centers recruiting patients (e.g., nephrology vs. rheumatology). These factors collectively contribute to the variability in study outcomes observed across different research efforts. Important answers to questions about the prognostic value of ANCA’s baseline status were eventually provided by an analysis of a large population from the French Vasculitis Study Group registry. The study unequivocally demonstrated that while the group of patients with seronegative AAV did not differ in terms of 5- or 10-year relapse-free rates from the group of GPA and MPA patients with antibodies, a detailed analysis of PR3-ANCA-positive patients showed a higher frequency of vasculitis exacerbations compared to the rest of the AAV population. It is important to note, however, that this relationship did not impact differences in overall survival [43]. Summarizing, it can therefore be concluded that not the presence of ANCA antibodies but their specificity may be helpful in stratifying the risk of relapse in patients who achieve AAV remission. Together with phenotypic analysis, it may in the future be a tool for classifying patients with individual clusters in terms of the risk of recurrence and be the basis for differentiating the treatment model in individual groups. The impact of such changes on overall survival would need to be verified but could be helpful in reducing the immunosuppressant’s exposure.

### 3.2. Cell Count, Genetics, Infections, Histopathology and Other Predictors at Baseline

Flow cytometry studies also allowed us to identify cell populations that may predict resistance to induction treatment. These include the population of CD27+CD38hi B cells, an increased percentage of which in the relapsers group was observed in both peripheral blood and urine. Their presence in the study material was interpreted as the result of migration from the kidneys, as these cells were also observed in the kidney biopsies of patients in the active phase of the disease [44]. In 2015, Graysson et al. conducted a trial focusing on the hypothesis that low-density granulocytes (LDGs) contribute to gene expression signatures in AAV. The RNA sequencing of whole blood in patients with AAV was compared. Differential expression between responders (n = 77) and nonresponders (n = 35) was detected in 2346 transcripts at the baseline visit. The increased expression of a granulocyte gene signature was associated with disease activity and decreased response to treatment. The source of this signature was likely LDGs [45].

Genome-wide association studies in AAV have led to the identification of the genetic background of AAV susceptibility such as *CTLA-4*, *PTPN22*, *FCGR2A*, *SERPINA1* (an endogenous inhibitor of PR3), and TLR9 genetic variations. It was also proven that GPA is associated with *HLA-DP1*, MPA with *HLA-DQ*, and EGPA with *HLA-DRB4* [46]. This discovery aroused questions about potential treatment resistance predictors among genetic factors. The carriers of *TNFSF13B rs3759467* (a single-nucleotide polymorphism) and homozygous carriers of *HLA-DPB1*04:01* were found to relapse earlier than non-carriers [47,48,49]. As genetic testing is not available in every clinical center and is also relatively complex and expensive, these research results seem to be more informative than have presently a practical impact.

The role of infectious factors on AAV development and course was debated for a long time. Thirty years ago, it was proven that the chronic nasal carriage of *S. aureus* identifies a subgroup of GPA patients who are at risk of relapses of the disease, suggesting a role for *S. aureus* in its pathophysiology and a possible clue for treatment [50,51]. This early observations were also supported by the fact that treating GPA patients with trimethoptim–sulfamethoxazole reduced the incidence of relapses [52]. An explanation of this phenomenon is complex and still not well described, but it is worth noting that GPA exhibits molecular features that allow its differentiation from other inflammatory disorders with nasal involvement and that a mucosal disbalance might orchestrate primary inflammation on site [53,54]. It has also been shown that in mice models, staphylococcus aureus plasmid-encoded 6-phosphogluconate dehydrogenase can induce anti-MPO autoimmunity, pointing at a possible molecular mimicry mechanism [55].

Most studies validating the histopathological classification of ANCA-associated glomerulonephritis focused on renal survival, although renal relapse probability was separately analyzed in a pooled cohort of patients from MEPEX and CYCAZAREM trials. Both clinical and histological parameters were taken under investigation. In this trial, sclerotic class ANCA-associated glomerulonephritis and the lack of interstitial infiltrates in baseline renal biopsies increased the instantaneous risk of renal relapse [56,57,58,59].

Some other relapse predictors at AAV flare have also been proposed: CCL18 serum levels, calprotectin at month 2 and 6 after RTX administration and a low number of circulating endothelial progenitor cells (EPCs) [60,61,62].

## 4. Relapse Predictors on Remission

Although cheap and easy to obtain, simple biomarkers of inflammation, like ESR, CRP, IL-6, IL-8, IL-15, IL-18BP, and matrix metalloproteinase-3 [MMP-3], seem to have a correlation with AAV activity and fail to predict relapse in longitudinal observations [63,64]. It raises questions about other possible tools for monitoring remission in order to early detect upcoming relapse.

### 4.1. ANCA Testing

Antibodies against PR3 most probably induce inflammation by directing the cells they bind to towards apoptosis, whereas antibodies against MPO stimulate cells to produce reactive oxygen species [65]. In vivo experiments have subsequently demonstrated that anti-MPO antibodies alone have the ability to induce full-blown vasculitis, whereas anti-PR3 antibodies have this potential only under specific conditions and typically induce milder forms of the disease [66,67,68,69,70]. These observations led to the hypothesis that an increase in ANCA titer may herald impending vasculitis exacerbations in patients who have achieved clinical remission, particularly in those with dominant capillaritis rather than granulomatous inflammation. Such measurements are cost-effective and non-invasive tools for monitoring patients in outpatient settings [71]. The initial reports on the predictive role of serial ANCA measurements in patients in remission suggested their high specificity. In a study by Tervaert et al., all exacerbations observed in the cohort of patients were preceded by a significant rise in ANCA titers [72]. In 1999, a serial analysis of ANCA levels in a group of patients with AAV showed that a significant titer increase was observed in 73% of exacerbations in patients with anti-MPO antibodies but only in 33% of exacerbations in those with anti-PR3 antibodies. This supports the hypothesis that the role of ANCA in monitoring remission in patients with AAV may vary depending on the disease phenotype [32]. Such observations were confirmed by subsequent studies. In a cohort of 166 patients with AAV, serial measurements of ANCA antibodies over a mean follow-up time of 49 ± 33 months revealed that a significant titer increase predicted vasculitis exacerbation primarily in patients with renal involvement (hazard ratio [HR], 11.09) and to a much lesser extent in patients with the involvement of other organs (HR, 2.79]) [73]. Additional information came from an extended analysis of the RAVE study (rituximab versus cyclophosphamide for remission induction), in which a rise in anti-PR3 antibodies was monitored. An increase measured by direct ELISA was associated with subsequent severe flare, especially in cases with renal involvement (HR, 7.94) and alveolar hemorrhage (HR, 24.19). This observation occurred only in the group of patients treated with rituximab, indicating that monitoring anti-PR3 levels may be applicable in a selected group of patients. It is also worth noting that a better predictive value was observed using the ELISA direct method than ELISA capture [74].

### 4.2. B-Cell Reconstitution

Over the past 15 years, rituximab has been increasingly playing a significant role in the treatment of AAV. As an antibody targeting B lymphocytes, it has not only proven to be non-inferior to conventional remission-inducing therapies but has also solidified its advantage as a maintenance therapy for sustaining remission [75,76,77,78]. It brought attention to B lymphocyte count as a predictive tool on remission. In the analysis of disease relapse during the first 6 months of remission induction therapy in the RAVE study, no correlation was observed between an increase in B cell count and subsequent exacerbation. None of the patients experiencing disease relapse within the first 6 months in the RTX-treated group had detectable B cells. Conversely, 47 out of 197 (24%) patients in the study had peripheral B cells present for up to 6 months of remission induction therapy but did not experience vasculitis relapse [39]. However, this observation changed from 6 months to 18 months of treatment where disease relapses were rare in the group of patients with B cell depletion [79]. In a long follow-up of RTX treatment, 71% of patients experienced B cell return at the time of relapse [80]. These observations were continued in the MAINRITSAN2 study. It compared two maintenance treatment regimens using rituximab for sustaining remission: fixed-dose maintenance (500 mg every 6 months) versus personalized dosing based on ANCA seroconversion, a 2-fold increase in ANCA titers, or the return of the peripheral B lymphocyte population. Both groups did not differ significantly in terms of relapse rates, but the personalized rituximab dosing group had reduced exposure to immunosuppression. Therefore, it can be inferred that these markers are useful for monitoring patients in remission to optimize treatment outcomes and cost-effectiveness [81]. Rituximab causes depletion not only of effector B lymphocytes but also of regulatory B cells. In a 2015 study by Bunch et al., the immunophenotypic profile of this cell lineage was observed, indicating that patients in remission of AAV who experienced a decrease in CD5+ regulatory B cells to less than 30% of all B lymphocytes tended to experience vasculitis exacerbations more quickly compared to those with CD5+ above 30% (median = 16 months versus 23 months, respectively) [82]. Other interesting observations focused on different B cell subtypes during remission. Naïve B cell repopulation appears to play a protective role against exacerbations, while an increase in PR3+ plasmablast populations signals exacerbation in patients treated with rituximab for remission induction [83]. Therefore, the immunophenotypic analysis of total B cell numbers appears to be an alternative or complementary approach to measuring ANCA levels as a potential predictor of exacerbation.

### 4.3. Other Potential Markers

Another proposed relapse predictor upon remission was a myelopoiesis gene signature in circulating leukocytes. In fact, PR3/MPO mRNA levels were not associated with future relapses in the analyzed cohort, but the authors hypothesized that undergoing steroid therapy might have affected the results [84]. Among other potential mechanisms in AAV etiopathogenesis, altered IgG glyco- and galactosialylation levels have also been debated. Those changes in the Fc region have been found to have an impact on IgG effector functions [85]. It was first described that in mass spectroscopy, low galactosylation and sialyation in total IgG1 but not PR3-ANCA IgG1 predicts disease reactivation in patients with GPA [86]. It was later shown by Wojcik et al. in a longitudinal observation that PR3-ANCA patients who experienced an ANCA rise and relapsed shortly thereafter presented lower IgG Fc-fucosylation levels compared to non-relapsing patients already 9 months before relapse [87]. Although the cost of such testing is relatively high, it is worth noting that the test was only performed as the second step in patients who experienced ANCA rise during follow-up. All these above mentioned methods are difficult to perform in the setting of everyday clinical practice.

Among other possible markers of upcoming flare, it is worth mentioning some simple and relatively cheap ones like urinary CD4+ count and persisting hematuria [88,89].

There are some combined relapse-predicting scores used to evaluate the treatment failures proposed, with the REACT score as the most recent one, and it is worth discussing the possibility of adding it to the baseline evaluation while starting treatment [90,91].

## 5. Discussion

All discussed markers are presented in Table 1. 

A series of analyses of potential relapse predictors in remission patients discussed in the literature prompts a consideration of whether using any of these predictors individually or in combination could justify intensifying treatment or if the optimal practice relies on an observation of clinical symptoms of disease relapse followed by implementing more aggressive therapy. It remains unclear how frequently follow-up visits should occur for patients exhibiting only serologic markers indicating impending exacerbation and whether the delayed recognition of subtly symptomatic disease recurrence may lead to increased chronic organ damage and consequently poorer outcomes. Additionally, it is unclear how often such markers should be measured and how to establish a cutoff for significant increases without raising false-positive prediction rates. It is worth noting that none of the analyzed potential relapse predictors demonstrate 100% specificity, and certainly, not every patient showing changes in these markers will develop vasculitis exacerbation or will do so with significant delay. It is also reasonable to evaluate potential differential diagnoses, like IgG4-related disease or malignancy, and overlapping syndromes, mainly treatment complications when early relapse occurs, especially in seronegative patients. The most challenging symptoms are those of the respiratory tract in patients with ENT involvement as infections might mimic disease progression or they coexist. It is still not well explored which of the discussed tools are disease-specific and distinguish between these two conditions.

## 6. Conclusions

Taking into consideration the heterogeneity of AAV, we can define, on the one hand, a subgroup with a low risk of relapse (female, anti-PR3-negative, high creatinine, no ENT involvement) who will be candidates for withdrawing the maintenance treatment, and on the other hand, the phenotype requiring probably lifelong maintenance therapy (male, PR3+, ENT, skin, eye involvement, low creatinine). But the majority of patients’ risk will be in between and will require real-time monitoring using a clinical observation, self-reporting and laboratory parameters monitoring to decide about maintenance therapy duration or intensification. Patients for whom the search for potential relapse markers has yielded less significant results so far include seronegative patients, the EGPA group, those with non-renal forms, and those treated with medications other than anti-CD19. There are several potential biomarkers proposed in the literature to evaluate relapse potential both at flare and upon the remission of AAV and they will need additional validation.

## Figures and Tables

**Table 1 diagnostics-14-01849-t001:** AAV relapse predictors.

Relapse Predictors	Well Described	Proposed
At baseline	antiPR3 presence	CD27+CD38hi B cells
		low-density granulocytes
		TNFSF13B rs3759467 (snp)
	respiratory tract and cardiovascular involvement	HLA-DPB1*04:01
		CCL18
		calprotectin
	*S. aureus* carriage	endothelial progenitor cells (EPCs)
		sclerotic class glomerulonephritis lack of interstitial infiltrates on renal biopsy
Upon remission	ANCA reappearance	galactosylation and sialyation in IgG1
		IgG Fc-fucosylation
	B cell reconstitution	urinary CD4+ count
		persisting hematuria

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
