# Peer review of "Relapse Predictors in Antineutrophil Cytoplasmic Antibody (ANCA)-Associated Vasculitis"

_diagnostics, 2024, doi:10.3390/diagnostics14171849_

Round 1

Reviewer 1 Report

Comments and Suggestions for Authors

Antineutrophilic cytoplasmic antibody (ANCA) associated vasculitides are a heterogeneous group of rare autoimmune conditions that causes an inflammation of blood vessels with various manifestations. To establish the diagnosis, a combination of clinical assessment with serological testing is needed, and a tissue biopsy many times confirms the diagnosis. Treatment for antineutrophilic cytoplasmic antibody (ANCA) vasculitides starts with induction of remission to avoid or slow down organ involvement. The definition of remission has been standardized by the European Vasculitis Society/European League against Rheumatism (EUVAS/EULAR) group. Predicting relapses remains pivotal for patient care, as relapses are associated with considerable morbidity caused by both disease- and therapy-related damage. So far, no biomarker for the prediction of relapse has been found to be reliable in GPA patients. Combined data of multiple studies has shown that ANCA titers can predict future relapses only to a limited extent. Prognosis and timely detection of recurrence of ANCA vasculitis is an urgent problem of modern medicine. From this point of view, the present review may be useful for the readers of the Journal Diagnostics. However, I do have a few comments:

(1) After the Introduction section, it is advisable to briefly outline the methodology of the review, including the type of review (narrative, systematic, or other), the databases used, the principles of inclusion and exclusion of publications, and the limitations.

(2) Obviously, readers of the article will be interested in the section on the differential diagnosis of ANCA vasculitis.

(3) The Discussion section is not detailed enough. Probably, the authors could have summarized the results of the study in this section, e.g., integrate the ANCA vasculitis markers in a table or systematize them in some other way. In this case, markers can be classified into several functional groups.In addition, the unresolved problems of prognosis and diagnosis of relapses in this disease could be discussed in this section.

Author Response

Comments 1: After the Introduction section, it is advisable to briefly outline the methodology of the review, including the type of review (narrative, systematic, or other), the databases used, the principles of inclusion and exclusion of publications, and the limitations.

Response 1: Thank you very much for your suggestion. I have added brief methodology description to the manuscript.

Comments 2: Obviously, readers of the article will be interested in the section on the differential diagnosis of ANCA vasculitis.

Response 2: Thank you very much for your suggestion We have added a note about potential differential diagnosis of an exacerbation to the manuscript.

Comments 3: The Discussion section is not detailed enough. Probably, the authors could have summarized the results of the study in this section, e.g., integrate the ANCA vasculitis markers in a table or systematize them in some other way. In this case, markers can be classified into several functional groups.In addition, the unresolved problems of prognosis and diagnosis of relapses in this disease could be discussed in this section.

Response 3: Thank you very much for pointing this out. We have added a table and some additional thoughts of unresoved problems to the discussion section of the manuscript.

Reviewer 2 Report

Comments and Suggestions for Authors

Comments: Authors addressed an important question in the review but manuscript require attention on following comments

1.      There are many recent and topic associated references/meta-analysis studies that can be incorporated in the manuscript.

2.      Limitations of ANCA testing must be discussed with appropriate references as ANCA sero-negativity is reported in many patients with GPA, MPA and EGPA.

3.      Section 2.2 is not properly segregated and other baseline predictors should be defined adequately. How RNA sequencing based 2,346 transcripts can be used in clinical set-up as predictors. Practically such testings are not validated in clinical set-up. Authors should discuss the limitations of each methods/predictors. Section title can be modified. Flowcytometry, GWAS, sequencing based tests are not validated in clinical set-up and are not economic.

4.      Relapse rates in recent ANCA-associated vasculitis trials should be discussed. Trials showing relapse may be associated with therapy switch and it may alter ANCA profile. NORAM, CYCAZAREM, CYCLOPS and MEPEX are discussed. Review should address this aspect of relapse. Other related trials can be discussed for e.g. IMPROVE, MAINRITSAN, RITUXVAS, WEGENT, RAVE.

Comments on the Quality of English Language

Minor correction may require.

Author Response

Comments 1: There are many recent and topic associated references/meta-analysis studies that can be incorporated in the manuscript.

Response 1: Thank you very much for this comment. Primarly, I havent’t added all the available data as the manuscript is already long and many publications were consistent with the data already presented but I find your suggestion very important and I have added some additional citations now.

Comments 2: Limitations of ANCA testing must be discussed with appropriate references as ANCA sero-negativity is reported in many patients with GPA, MPA and EGPA.

Response 2: Limitations of ANCA testing is an important issue and are presented in the manuscript in citations number 25-30

Comments 3: Section 2.2 is not properly segregated and other baseline predictors should be defined adequately. How RNA sequencing based 2,346 transcripts can be used in clinical set-up as predictors. Practically such testings are not validated in clinical set-up. Authors should discuss the limitations of each methods/predictors. Section title can be modified. Flowcytometry, GWAS, sequencing based tests are not validated in clinical set-up and are not economic.

Response 3: Thank you very much for this comment, I agree that these methods are not practical. I wanted to mention some interesting, non-clinical predictors that were analyzed. I have highlighted proper limitations of these methods in red in the manuscript.

Comments 4: Relapse rates in recent ANCA-associated vasculitis trials should be discussed. Trials showing relapse may be associated with therapy switch and it may alter ANCA profile. NORAM, CYCAZAREM, CYCLOPS and MEPEX are discussed. Review should address this aspect of relapse. Other related trials can be discussed for e.g. IMPROVE, MAINRITSAN, RITUXVAS, WEGENT, RAVE.

Response 4: Trials have delivered most of the valuable data in this topic. Thank you very much for your suggestion. The data from the trials suggested are present in the manuscript even though the trials are not named (RAVE trial- citation number 76, RITUXVAS trial- number 75, WEGENT trial- number 34). IMPROVE trial did not show any differences in organ demonstrations between the two groups that relapsed and because of that it wasn’t mentioned. MAINRITSAN trial relapsers anaylsis results were consistent with other ANCA-specificity result in this field so it wasn’t presented.